# Associations between Racing Thoroughbred Movement Asymmetries and Racing and Training Direction

**DOI:** 10.3390/ani14071086

**Published:** 2024-04-03

**Authors:** Bronte Forbes, Winnie Ho, Rebecca S. V. Parkes, Maria Fernanda Sepulveda Caviedes, Thilo Pfau, Daniel R. Martel

**Affiliations:** 1Hong Kong Jockey Club, Hong Kong, China; bronte.s.forbes@hkjc.org.hk (B.F.); winnie.ho@cityu.edu.hk (W.H.); 2Singapore Turf Club, Singapore 738078, Singapore; 3Department of Veterinary Clinical Sciences, City University of Hong Kong, Hong Kong, China; rparkes@sgu.edu; 4Department of Veterinary Clinical Sciences, Royal Veterinary College, London NW1 0TU, UK; fernanda@gvgbrooksequine.co.uk; 5Faculty of Kinesiology, University of Calgary, Calgary, AB T2N 1N4, Canada; daniel.martel@ucalgary.ca; 6Faculty of Veterinary Medicine, University of Calgary, Calgary, AB T2N 1N4, Canada

**Keywords:** movement symmetry, Thoroughbred, kinematics, stride segmentation, inertial sensors, gait

## Abstract

**Simple Summary:**

Racehorses race and train in either a clockwise or anticlockwise direction and may adapt how they move accordingly, potentially creating asymmetry of movement between their left and right sides. Wireless inertial sensors measured the vertical head and pelvic movement of 307 Thoroughbreds (156 from the anticlockwise exercising Singapore turf club, “STC”, and 151 from the clockwise exercising Hong Kong Jockey Club, “HKJC”) trotting in a straight line in-hand on a firm surface. Seven vertical movement symmetry variables comparing the vertical movement between the left- and right-side halves were extracted, and statistical analyses were conducted to compare the number of left and right asymmetrical horses between cohorts, for both the fore- and hindlimbs. Our results revealed that there were significantly more left forelimb asymmetrical horses in the HKJC cohort compared to the STC cohort. Additional analyses revealed significant differences between cohorts, with the HKJC horses on average showing higher levels of left fore- and hindlimb asymmetry and, for the STC horses, right fore- and hindlimb asymmetry. This suggests that horses may indeed adapt their movement to favor one side over the other based on the direction they typically train and race in.

**Abstract:**

Background: Racehorses commonly train and race in one direction, which may result in gait asymmetries. This study quantified gait symmetry in two cohorts of Thoroughbreds differing in their predominant exercising direction; we hypothesized that there would be significant differences in the direction of asymmetry between cohorts. Methods: 307 Thoroughbreds (156 from Singapore Turf Club (STC)—anticlockwise; 151 from Hong Kong Jockey Club (HKJC)—clockwise) were assessed during a straight-line, in-hand trot on firm ground with inertial sensors on their head and pelvis quantifying differences between the minima, maxima, upward movement amplitudes (MinDiff, MaxDiff, UpDiff), and hip hike (HHD). The presence of asymmetry (≥5 mm) was assessed for each variable. Chi-Squared tests identified differences in the number of horses with left/right-sided movement asymmetry between cohorts and mixed model analyses evaluated differences in the movement symmetry values. Results: HKJC had significantly more left forelimb asymmetrical horses (Head: MinDiff *p* < 0.0001, MaxDiff *p* < 0.03, UpDiff *p* < 0.01) than STC. Pelvis MinDiff (*p* = 0.010) and UpDiff (*p* = 0.021), and head MinDiff (*p* = 0.006) and UpDiff (*p* = 0.017) values were significantly different between cohorts; HKJC mean values indicated left fore- and hindlimb asymmetry, and STC mean values indicated right fore- and hindlimb asymmetry. Conclusion: the asymmetry differences between cohorts suggest that horses may adapt their gait to their racing direction, with kinematics reflecting reduced ‘outside’ fore- and hindlimb loading.

## 1. Introduction

The direction of training and racing, clockwise or anticlockwise, varies between Thoroughbred racing jurisdictions globally. However, individual jurisdictions will commonly have a predominant direction that their horses exercise and compete in [1,2,3]. When horses exercise in a circle, the forces applied to the inside and outside limbs differ, so it is feasible that horses exercising in a predominant direction may adapt to develop asymmetries of movement between their left and right sides [4,5,6,7]. The use of objective measurements of the symmetry of movement in horses is increasing globally, with the information obtained allowing for the more empirical sharing of information between veterinarians [8,9]. With increased prize money and improved accessibility for international horse movement, the numbers of horses travelling to other jurisdictions to race are also rising. These internationally travelling horses will commonly compete in a direction opposite to what they would predominantly exercise in at their own jurisdiction [10]. Therefore, having knowledge of any possible movement symmetry biases that may occur with horses racing and training in a predominant direction will be useful both when assessing horses objectively for movement symmetry, so that any changes in movement patterns can be accounted for when interpreting objective asymmetry measurement results, and also for developing training programs in horses that may potentially travel to compete in a jurisdiction that races in the opposite direction.

The purpose of this study was to assess and quantify any differences in the symmetry of movement between horses training and racing in different directions through the use of two cohorts of Thoroughbred racehorses predominantly exercising in different directions in Singapore (anticlockwise) and Hong Kong (clockwise). Having this knowledge will potentially allow for the modification of the training programs of horses intending to compete in jurisdictions that race in an opposite direction to their predominant one. Adapting to training may allow for horses to adapt their musculoskeletal systems to the different forces that occur to the inside and outside limbs when exercising in a circle. Knowledge of any potential differences in the symmetry between sides in horses that are trained in a predominant direction will also assist veterinarians when assessing the potential significance of asymmetries in horses that are being assessed objectively.

The aim of this study was to quantify the gait asymmetry in two cohorts of Thoroughbreds differing in their race training direction. Specifically, this study compares the direction that horses predominantly trained and raced in and the number of horses with gait asymmetries present between a cohort of anticlockwise-training horses (from the Singapore Turf Club) and a cohort of clockwise-training horses (Hong Kong Jockey Club). Additionally, the magnitude and relative direction of pelvis and poll asymmetry measures were compared between cohorts. It was hypothesized that there would be a significant difference in the direction of asymmetry between the cohorts, with an opposing dominant asymmetry direction.

## 2. Materials and Methods

### 2.1. Sample

Towards achieving the goal of quantifying the effect of racing and training direction on gait kinematics and gait asymmetries, two clubs of opposing and exclusive racing and high-intensity training direction, with otherwise equivalent training and racing facilities, were included in this study, which were the anticlockwise Singapore Turf Club (STC) in Singapore, and the clockwise Hong Kong Jockey Club (HKJC) in Hong Kong (see Table 1 for details on the tracks present at both racing facilities). A total of 522 Thoroughbred horses in race training were included in this study, with 286 horses training at STC, and 236 horses in training at HKJC. All horses from both the HKJC and STC cohorts were Thoroughbreds within the standard breed height range of 15 to 16.1 hands (range = 152.4–163.6 cm). STC horses comprised of 249 geldings, 6 entire males, 28 females, and 3 unknown, with the HKJC having 235 geldings and 1 entire male. Both cohorts were shod in all four limbs with the majority being with aluminum shoes. After the introduction of an exclusion criterion to remove horses with large asymmetry measures (≥25 mm), a sample of 307 horses remained (156 from STC, 151 from HKJC); this threshold was selected as a previous study demonstrated that 95% of weekly variations for head-related asymmetry measures obtained from inertial sensors are between 22 and 26 mm [11]. HKJC horses included had a mean (SD) body mass of 512.5 (31.2) kg with a range of 469 to 575 kg, while the median (IQR) age (to the nearest year) was 5 (2) years with a range from 3 years to 10 years. STC horses included had a mean (SD) body mass of 500.7 (32.8) kg with a range from 412 kg to 590 kg and a median (IQR) age of 4.2 (1.6) years and range of 2.3 to 8.7 years. The experimental protocol for this study was approved by the following institutional review boards and ethics review committees: the Royal Veterinary College’s Ethics and Welfare Committee (URN 2013 1238), the City University of Hong Kong’s Ethics Review Committee (ERC/040/2021), and the University of Calgary Animal Care and Use Committee (AC21-0185).

### 2.2. Data Collection

Horses were assessed trotting in a straight line in-hand with a halter and Chifney, led from the left side, on a firm surface at their respective clubs. At STC, the surface used during the trials was a flat asphalt surface. At the HKJC, the surface used during the trials was a firm rubber surface with a concrete underlay for 40% of the horses, with the other 60% of horses trotting on a flat concrete surface. All trials were conducted by a trainer (or stable hand) instructed to lead horses during two 20 m trot-up runs in a straight line with a loose rein, at their preferred speed. An observer was present during the trials, with one run being directed away from the observer, and the other being directed towards the observer. If the horse was fractious, the procedure was repeated until two consistent runs were achieved. Inertial measurement units (STC: MTw, XSens, HKJC: Xsens DOT) were placed on the pelvis, midline dorsal to the tuber sacrale (sacrum, hindlimb measures), left and right tuber coxae, placed equidistant from the midline sacrum sensor, and head (poll, forelimb measures) of each horse. At the Singapore Turf Club, IMU sensor data were communicated via a proprietary protocol (XSens, Awinda, Movella, Henderson, NV, USA) to a nearby laptop computer. At Hong Kong Jockey Club, data were stored onboard the sensors during data collection and then transmitted via Bluetooth (Bluetooth LE) to a smartphone (Apple iPhone12, Apple Inc., Cupertino, CA, USA) before then being uploaded to a Windows-based (Microsoft, Redmond, WA, USA) computer for data analysis with custom-written software (MATLAB R2022b, Natick, MA, USA). For both systems, data were collected with a sampling rate of 60 Hz sufficient for the purpose of calculating upper body movement symmetry in trotting horses [12]. For the Singapore Turf Club horses, data were collected between November 2014 and May 2016 and for the Hong Kong Jockey Club horses, the data were collected between 2016 and 2023.

### 2.3. Data Processing

IMU data were processed using previously published methods [13,14]. Continuous IMU data streams were segmented into individual strides using a method adapted from Starke and colleagues [15]. For each sensor location, asymmetry was quantified by calculating differences between vertical displacement minima (MinDiff), maxima (MaxDiff) and upward (UpDiff) amplitudes, and hip hike difference (HHD) for each identified stride cycle; the median of each of these variables was calculated for each horse. Negative values indicate a left asymmetry (reduced load on the left limb of a pair), and positive values indicate a right asymmetry (reduced load on the right limb of a pair). Presence and direction of asymmetry was then identified for each of the variables as displacement ≥5 mm; horses with no identified asymmetry or with any median displacement variable ≥25 mm were excluded from analysis.

### 2.4. Statistical Analyses

To identify differences in the number of horses with left- or right-sided movement asymmetry between cohorts (HKJC compared to STC) and within cohorts, Chi Squared Tests were conducted to compare the number of horses with identifiable asymmetry (the asymmetry displacement variable ≥ 5 mm) for each of the asymmetry displacement variables. Shapiro–Wilks tests of normality were conducted on each asymmetry displacement variable. Mixed model analyses (cohort as Fixed effect, horse as Random effect) were conducted to evaluate differences in the values of each variable between the cohorts, allowing for a quantification of the relative asymmetry differences of the variables. All statistical analyses were conducted at an α level of 0.05 and performed with SPSS (IBM SPSS Statistics 25, IBM, Armonk, NY, USA).

## 3. Results

The sample used for the analyses totaled 307 horses, which included 156 from STC and 151 from HKJC. Whole-sample and cohort-specific means for the displacement difference variables can be found in Table 2. Numbers of left and right asymmetrical horses presented by the displacement variable and by cohort are shown in Table 3 and Table 4. For the sample as a whole, Shapiro–Wilks tests revealed that the displacement difference variables MinDiff and UpDiff for both the pelvis and poll, as well pelvis HHD, were not normally distributed (*p* < 0.05). Residuals were assessed during the mixed model analyses, finding that the residuals were approximately normally distributed for all variables.

### 3.1. Number of Left and Right Asymmetrical Horses

Chi Squared tests comparing the number of horses with asymmetries between cohorts (HKJC—STC) identified by each displacement difference variable for both right and left asymmetries at the head revealed that HKJC had significantly more horses with left forelimb asymmetry (Head: MinDiff *p* < 0.0001, MaxDiff *p* < 0.03, UpDiff *p* < 0.01) compared to STC (Figure 1). There were no significant differences in the pelvis-related variable Chi Squared tests (*p* > 0.05) (Figure 2). The results of the between-cohort Chi Squared tests are shown in Table 3. Chi Squared comparing the number of left and right (left–right) asymmetries within the cohorts identified by each displacement difference variable at the head revealed that STC had significantly more right forelimb asymmetrical horses than left (Head: MinDiff *p* < 0.001, UpDiff *p* < 0.01). Chi Squared tests of the pelvis revealed that STC had significantly more right hindlimb asymmetrical horses than left (Pelvis: MinDiff *p* < 0.02). The results of the within-cohort Chi Squared tests are shown in Table 4.

### 3.2. Magnitude of Left and Right Asymmetrical Movement Symmetries

Mixed model analyses revealed significant differences in the displacement differences for both the pelvis and head variables. There was a significant difference in the pelvis MinDiff (F [1305] = 6.732, *p* = 0.010) between cohorts, with a mean difference (SE) of 2.0 (0.8) mm (95% CI −3.5:−0.5). Similarly, the pelvis UpDiff was also significantly different between cohorts (F [1305] = 5.371, *p* = 0.021), with a mean difference (SE) of −2.8 (1.2) mm (95% CI −5.1:−0.4). The head MinDiff (F [1305] = 7.638, *p* = 0.006) and UpDiff (F [1305] = 5.794, *p* = 0.017) were significantly different between the cohorts, with mean differences (SE) of −3.0 (1.1) (95% CI −5.2:−0.9) and −3.5 (1.5) mm (95% CI −6.4:−0.6), respectively (Table 2). For all significant differences, the HKJC mean values were negative (left asymmetry), and the STC mean values were positive (right asymmetry) (Figure 3).

## 4. Discussion

The present study was designed to investigate whether there are any predilections to how the asymmetries of movement when trotting in a straight line in-hand may occur when horses are exercised in a predominant direction. This was achieved by comparing various movement symmetry indices (MinDiff, MaxDiff, UpDiff for the head and pelvis, and HHD for the pelvis) between horses from two jurisdictions that predominantly race and train at high intensities in opposing directions—Singapore in an anticlockwise direction (156 horses out of 286 horses) and Hong Kong in a clockwise direction (151 horses out of 236 horses) using a validated objective symmetry analysis system whilst trotting in-hand on a firm surface [13,14]. Differences in asymmetry, although relatively small, where found for both forelimb (head MinDiff and head UpDiff) and hindlimb (pelvis MinDiff and pelvis UpDiff) indices with a predilection for a side based on the predominant direction of intense exercise.

Although the values determined to be significantly different for the displacement indices measured were small between the clockwise cohort from Hong Kong and the Singapore cohort that trained and raced in a predominantly anticlockwise direction, given the large number of stride cycles that Thoroughbred horses undertake in training, it is feasible that the accumulation of the effect that these small differences have on the total force on the limb over time may still be important to consider [12,16,17,18]. For values that showed a significant difference, the Hong Kong horses had negative mean values, suggesting a propensity for left asymmetry, i.e., an asymmetry indicating a reduced downward or upward movement during the stance phase of the (left) limb that is on the outside of the circle during clockwise training and racing. The Singapore horses, on the other hand, showed positive mean values, suggesting right asymmetry, again a movement pattern indicative of a reduced downward or upward movement during the stance phase of the (right) limb that is on the outside of the circle during anticlockwise training and racing [19,20]. Significant differences between cohorts were found in both the hindlimbs (Pelvis MinDiff and UpDiff) and forelimbs (Head MinDiff and UpDiff), though the magnitude of the differences were relatively small for all of the displacement variables compared to pre-established thresholds for lameness [21]. Regardless, these results support the hypothesis that exercising in a predominant direction affects the symmetry of movement of Thoroughbreds when they are assessed by trotting in-hand in a straight line. More specifically, these findings suggest that there is a propensity for horses to show movement patterns when trotting in-hand in a straight line that indicate a reduced force to the limb that would have been on the outside when exercising around a curve in a predominant direction. This appears contrary to the findings of previous research that report that the outside limb of horses exercising in a circular motion experience greater force at a gallop, so it would be expected that the asymmetry pattern trotting in a straight line would follow a similar pattern [4,5,6,7]. However, the movement pattern findings from this study indicate that there are differences in the force of the limb that would be on the outside during galloping on a curve to when the horse is trotting in a straight line, with less force produced in the limbs (front and hind) on the outside during a gallop than when assessing horses trotting in a straight line. A previous study on curve running [22] found that there is a higher duty factor for the inside limb in horses negotiating a bend when working at a gallop on the correct lead, and in humans, it has been suggested that a higher duty factor when running on a treadmill can reduce the peak force on limbs [23]. It has also been reported that peak vertical ground reaction forces are directly affected by a duty factor with an increase in the duty factor associated with a decrease in the peak vertical ground reaction force following the equation [24,25]:Fzmax = πpmg/4β

Fzmax = peak vertical ground reaction force; p = proportion of the mass of the horse carried by each limb of a contralateral pair [typically 0.6 for the front and 0.4 for hinds]; m = mass of the horse in kilograms; g = gravitational constant; and β= duty factor.

Our results suggest that the forces on the limbs are different during galloping on a curve and trotting in a straight line in horses that exercise using a curved track with a predominant direction. There are also changes over time with respect to the stride duration when running on a curve, indicating the presence of an adaptation with training [26]. It is possible that the differences observed in this study are the result of gait adaptations to racing and training in a predominant direction. With the knowledge that bone and soft tissue can adapt to changes in force [16,27,28,29], it is possible that horses racing and training in a predominant direction may undergo similar musculoskeletal adaptations [30,31,32,33]. However, as this was a cross-sectional study, we are unable to determine whether these differences are due to an adaptation due to training and exercising in a single direction. Like humans, animals, including horses, show handedness, or laterality, which indicates a predisposition to favor one side of the body compared to the other and this may be a learned process in horses allowing for the potential of being influenced by training [2,3,10,31,34,35,36]. Knowing if and how these various influences affect a horse’s measurable movement symmetry would be useful to take into consideration when using objective methods to assess a horse’s movement, for example, whilst undertaking lameness examinations, for the longitudinal monitoring of horses’ movement during a training regime, or when screening horses for their suitability to participate in racing or other equestrian competitions [37,38,39]. A recent study in >200 clinical cases [40] has provided the first evidence that when differentiating between sound and lame horses, different normalized threshold values may be needed for left and right lame horses, so including information on likely changes to asymmetry measures in horses that exercise at high intensities in a predominant direction should also be considered when developing thresholds of movement symmetry for Thoroughbreds in race training. Ultimately, knowledge about how any exercise- and direction-related conditions affect movement asymmetry may contribute evidence assisting with the design of specific training programs, for example, allowing for the adaptation of racing Thoroughbreds to different ways of moving during training and racing with the potential to reduce injury [2,10].

The loading on the outside and inside limbs can be affected by both the degree of camber on a bend and the gait at which the horse negotiates the banked curvature, as this affects the duty factor, so both the degree of track camber and speed of the horse negotiating the bend need to be considered when assessing the mechanical loading of the inside and outside limbs during exercise on an elliptical track [5,41,42]. Training surfaces in both Hong Kong and Singapore are banked at their corners so consideration of the potential affect that banking and camber may have on the symmetry of movement of the horses sampled in this study should be given. The training load (speed and distance) of each horse was not available for this study and the degree of camber of each track used for racing and training, although similar, was not identical between the two jurisdictions (Table 1). Although these factors have the potential to affect the amplitude of movement asymmetry due to their effect on the degree of force to both the inside and outside limbs, they are unlikely to change the side of asymmetry, which was the variable being measured for this study. With the exclusion criteria including horses with a threshold of 5 mm or less displacement, selected as this is the approximate margin of error in displacement based on the linear acceleration measured by these sensors from repeatability studies [17], or over 25 mm displacement, it is possible that the differences in the horse’s speed and degree of the tracks’ camber may have contributed to the exclusion decision of some horses.

An asphalt surface was used for most of the trot-up examinations of the horses in both Singapore and Hong Kong; however, a number of horses were sampled whilst trotting in-hand on a rubber-over-concrete surface for the HKJC cohort, and the effect of the trot-up surface on the results was not tested in this investigation. Previous work by Azevedo et al. (2015) showed that the impact or push-off lameness presentation for both front and hindlimbs are not altered by the type of track surface that horses trot on [43]. As all surfaces used for this study were considered firm, there was no anticipated effect of the surface type on the measures taken from the gait assessment between the STC and HKJC cohorts and within the HKJC cohort.

It should also be acknowledged that the daily training intensity is likely to have differed between horses both within and between the STC and HKJC cohorts. However, previous work by Sepulveda Caviedes et. al. (2017) on the repeatability of gait analysis showed that variability was similar between daily and weekly symmetry analysis assessments, suggesting that variability in the measures is consistent despite differences in training [11]. Therefore, this would likely have little to no effect on the results of this assessment, especially in relation to the side of the movement displacement changes.

Previous studies in racing greyhounds [44] and racing Thoroughbreds [45] have demonstrated morphological differences between sides in the skeletal system of animals racing in a predominant direction, which suggests that exercising at high intensities in a predominant direction may create similar movement asymmetries in other racing animals via the same adaptive processes. Given the small, although significant, overall differences in movement patterns measured across a range of horses with variations in multiple horse- and track-based factors, further investigation into the underlying processes that are creating the asymmetries and how the predominant direction of training and racing may affect other horse-based variables, such as, for example, a hoof shape associated with changes in limb angulation [5] and musculoskeletal adaptation, is warranted.

Findings from this study support the idea that exercising in a predominant direction develops measurable predictable asymmetries in horses trotting in-hand in a straight line. Allowing for musculoskeletal changes that may be important for the adaptive process for training and racing in a predominant direction has the potential to reduce injury rates in racing Thoroughbreds. This should, therefore, be considered when training horses that intend to compete in a jurisdiction that races in the opposite direction to their usual way. It might also be useful for veterinarians working in different racing jurisdictions to take this into consideration when using objective measures of movement symmetry during movement symmetry assessments of Thoroughbred horses for fitness to race, in particular, for horses traveling to compete in races undertaken in the direction opposite to their normal exercising pattern [22]. The results from this study were based on movement symmetry data from horses trotting in-hand in a straight line, so they do not provide direct evidence for the findings discovered during a gallop on a curved surface. However, previous research has noted that there are small differences in the speed and stride length at the gallop during racing and measurable changes to ground reaction forces that are both predictive of impending musculoskeletal injury in horses [46,47]. Further longitudinal studies are, therefore, warranted to identify optimal parameters for the prediction of impending injuries in racing Thoroughbreds at sub-maximal and maximal speeds, as well as investigate the potential physical and biological (performance, injury, and morphological adaptation) implications of these differing asymmetries identified in this work.

## 5. Conclusions

Significant differences in asymmetry between the fore- and hindlimbs of the two cohorts were noted, suggesting that horses may adapt their gait to a predominant direction of exercise and that this adaptation is reflected as a change in their movement pattern, indicating a reduced loading of the ‘outside’ fore- and hindlimbs when trotting in a straight line. This was demonstrated through an objective, quantitative IMU-based assessment of the horses’ movement when trotting in-hand, on a firm surface, in a straight line. Further research is warranted to determine the underlying processes that may be influencing these changes in the symmetry of movement in Thoroughbred racehorses training in a predominant direction.

## Figures and Tables

**Figure 1 animals-14-01086-f001:**
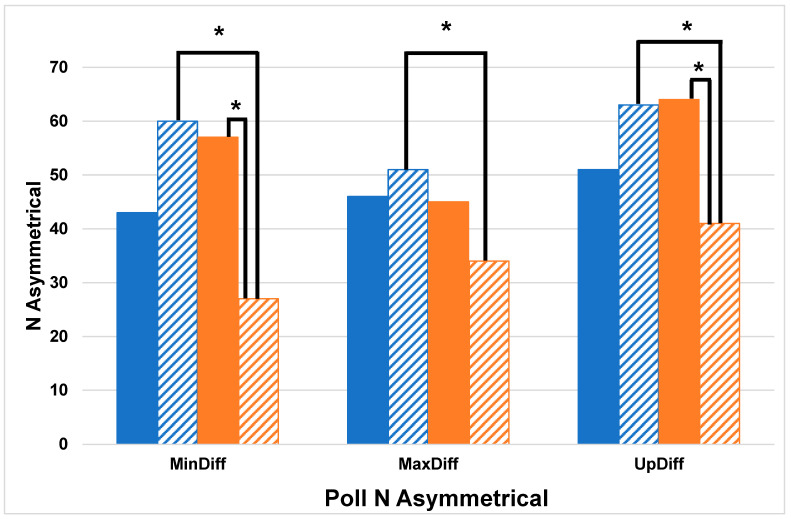
Count of asymmetrical horses identified from poll sensor asymmetry measures which included vertical displacement minimum difference (MinDiff), vertical displacement maximum difference (MaxDiff), and upward displacement amplitude difference (UpDiff) (blue = Hong Kong Jockey Club, HKJC, Orange = Singapore Turf Club, STC, Solid = Right, Hashed = Left), * denotes significance level of *p* < 0.05.

**Figure 2 animals-14-01086-f002:**
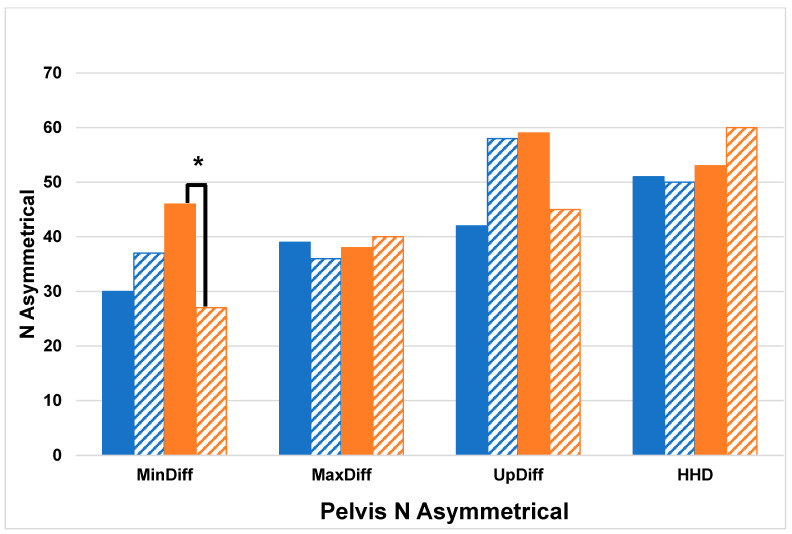
Count of asymmetrical horses identified from pelvis sensor asymmetry measures which included vertical displacement minimum difference (MinDiff), vertical displacement maximum difference (MaxDiff), an upward displacement amplitude difference (UpDiff), and hip hike difference (HHD) (blue = Hong Kong Jockey Club, HKJC, Orange = Singapore Turf Club, STC, Solid = Right, Hashed = Left), * denotes significance level of *p* < 0.05.

**Figure 3 animals-14-01086-f003:**
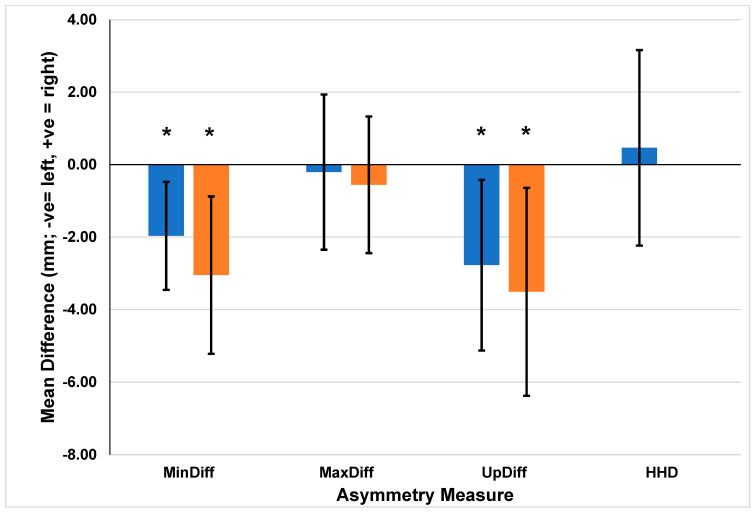
Mean asymmetry metric differences between cohorts for the asymmetry measures which included vertical displacement minimum difference (MinDiff), vertical displacement maximum (MaxDiff), upward displacement amplitude difference (UpDiff) for the head and pelvis, and hip hike difference (HHD) for the pelvis (Hong Kong Jockey Club−Singapore Turf Club; HKJC−STC) based on sensor location and asymmetry measure (blue = Pelvis, Orange = Poll), * denotes significance level of *p* < 0.05.

**Table 1 animals-14-01086-t001:** Circumference in meters and degree of maximum camber for the racing and training tracks at the Singapore Turf Club (STC) and the Hong Kong Jockey Club (HKJC) including the HKJC Happy Valley turf racing surface and the HKJC training facility in Conghua China (CRC).

Track	Circumference (Meters)	Home Straight Maximum Camber	First Corner Maximum Camber	Second Corner Maximum Camber	Back Straight Maximum Camber
**HKJC Turf**	1898	2.5%	4%	4%	2.5%
**HKJC Large All Weather**	1558	2.5%	3%	3%	2.5%
**HKJC Small All Weather**	1429	2.5%	3%	3%	2.5%
**HKJC Happy Valley Turf**	1417	5%	6.67%	5%	3.3%
**HKJC CRC Turf**	1952	1%	4%	5%	1%
**HKJC CRC Small All Weather**	1728	1.5%	3%	3%	1.5%
**HKJC CRC Large All Weather**	1592	1.5%	3%	3%	1.5%
**STC Turf**	2000	2.5%	6.5%	6.0%	2.5%
**STC Polytrack**	1550	2.5%	4.5%	4.5%	2.5%
**STC Sand Track 1**	1300	2.5%	6.0%	6.0%	2.5.0%
**STC Sand Track 2**	1650	1.5%	2.5%	N/A	N/A

**Table 2 animals-14-01086-t002:** Mean values for each displacement difference variable (in mm) for the whole sample, as well as for each individual cohort (HKJC = Hong Kong Jockey Club; STC = Singapore Turf Club).

Location	Variable	Whole Sample N = 307		HKJC N = 151		STC N = 156		HKJC—STC	
		Mean	SD	Mean	SD	Mean	SD	Mean Diff.	95% CI
Pelvis	MinDiff	0.7	6.7	−0.3	6.2	1.6	7.1	**−2.0 ***	**−0.5** **−3.5**
	MaxDiff	0.1	7.7	0.2	7.4	0.0	8.0	−0.2	1.9 −1.5
	UpDiff	0.3	10.6	−1.1	9.8	1.6	11.1	**−2.8 ***	**−0.4** **−5.1**
	HHD	0.0	12.0	0.3	11.1	−0.2	12.8	0.5	3.2 −2.2
Poll	MinDiff	0.8	9.8	−0.7	10.4	2.3	8.9	**−3.0 ***	**−0.9** **−5.2**
	MaxDiff	−0.1	8.4	−0.4	8.4	0.2	8.4	−0.6	1.3 −2.4
	UpDiff	0.8	12.9	−1.0	13.3	2.5	12.2	**−3.5 ***	**−0.6** **−6.4**

* denotes significance level of *p* < 0.05. Significant mean differences and 95% confidence intervals given in bold.

**Table 3 animals-14-01086-t003:** Side asymmetry comparison between cohorts: number of identified asymmetries for each displacement difference variable (Difference = Hong Kong Jockey Club—Singapore Turf Club, diff. = HKJC—STC).

Location	Variable	# of Left Asymmetrical			# of Right Asymmetrical		
		HKJC	STC	Diff.	HKJC	STC	Diff.
Pelvis	MinDiff	37	27	10	30	46	−16
	MaxDiff	36	40	−4	39	38	1
	UpDiff	58	45	13	42	59	−17
	HHD	50	60	−10	51	53	−2
Poll	MinDiff	60	27	**33 ***	43	57	−14
	MaxDiff	51	34	**17 ***	46	45	1
	UpDiff	63	41	**22 ***	51	64	−13

* denotes significance level of *p* < 0.05. Significant differences given in bold.

**Table 4 animals-14-01086-t004:** Side asymmetry comparison within cohorts (HKJC = Hong Kong Jockey Club; STC = Singapore Turf Club): number of identified asymmetries for each displacement difference variable (Difference = Left–Right).

Location	Variable	HKJC # of Asymmetrical			STC # of Asymmetrical		
		Left	Right	Diff.	Left	Right	Diff.
Pelvis	MinDiff	37	30	7	27	46	**−19 ***
	MaxDiff	36	39	−3	40	38	2
	UpDiff	58	42	16	45	59	−14
	HHD	50	51	−1	60	53	7
Poll	MinDiff	60	43	17	27	57	**−30 ***
	MaxDiff	51	46	5	34	45	−11
	UpDiff	63	51	12	41	64	**−23 ***

* denotes significance level of *p* < 0.05. Significant differences given in bold.

## Data Availability

A spreadsheet including the processed data (median displacement difference variable for each included horse, left/right asymmetry binary) is available at this figshare link https://doi.org/10.6084/m9.figshare.24312568.

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
