# Peer review of "Associations between Racing Thoroughbred Movement Asymmetries and Racing and Training Direction"

_animals, 2024, doi:10.3390/ani14071086_

Round 1
Reviewer 1 Report
Comments and Suggestions for Authors
This is an original, well-deigned and well-presented study that is of interest in both basic and applied biomechanics and may have relevance to orthopaedic injury in racehorses. I have no major concerns.
Specific comments
Presume all horses were led from their left-side - what impact does this have on assessment of gait symmetry? Does it have the potential to create any bias?
Were horses trotted in headcollars or bridles?
When were horses trotted up in relation to daily training? Were all horses trotted having done similar training the previous day? Presumably the previous days training intensity could affect symmetry?
HKJC - Did you test for any significant effect of (difference between) trot up surface? If not, why not and if so, please include results? I think before combining the two surfaces I would like to be assured that they were not sig different.
Can you confirm, and make explicit in the text, that all horses at each track were only trained in one direction. i.e. their were no horses included that had been trained in both directions?
In your statistical description, there is no mention of whether normality of distribution tests were undertaken as a pre-requisite for parametric analysis?
No mention of variance presentation - SD or SE? Presume SD as given in some results but variance missing for example in Figure 3.
No key in Figure 1 but key in figure 2?
Are these significant differences likely to be biologically relevant?
What is the likely relationship/influence of assymetry in a "symmetrical" gait at low speed (i.e. trot) to the biomechanics of a racing gallop? Would analysis of gait at racing gallop be expected to confirm the results at trot or might this effect be diminished due to speed, gait assymetry and endogenous analgesia i.e. endorphin release.
Reviewer 2 Report
Comments and Suggestions for Authors
Review
Associations between racing Thoroughbred movement asymmetries and racing and training direction.
REVIEWER: OVERALL IMPRESSIONS – the manuscript attempts to introduce the asymmetry of movement in Thoroughbred racehorses that race and train in one direction. There are previous studies on the subject not mentioned here. 1) femoral asymmetry in Thoroughbred racehorses racing and training clockwise. 2) greyhound racing fractures associative with one directional training. 3} standardbred racing (trotters). These could have been used to also support the findings in this study.
TITLE
Not necessary but would suggest something along the lines of - Associations between Asymmetrical Movement in Thoroughbred Racehorses to Directional Training and Racing.
SIMPLE SUMMARY
Here the authors mention left and right asymmetry, but do not clarify whether these relate to the forelimb and or hind limb. For the lay person, nominate which limbs are mostly affected by asymmetrical gait patterns and associate these to directional training.
ABSTRACT
Nominate which limbs are the most affected by asymmetrical gait patterns and associate these to directional training.
Line 44 – ‘outside’ limb loading – fore or hind limb?
KEY WORDS
Stride segmentation was not mentioned in the manuscript except in key words – segmented was mentioned only once. The mechanical beat of the gallop or trot are not mentioned, so ‘stride segmentation’ is superfluous here. Asymmetrical gait patterns might be better suited.
INTRODUCTION
Could site other studies that have performed similar research e.g. greyhounds and standardbreds for a comparative background. The idea of findings supporting a common issue associative to directional training/racing has merit, and especially when such horses finish a racing career and venture into the recreational disciplines such as dressage, eventing, and showjumping. Thus, the findings from the current study could imply that directional forces might compromise the musculoskeletal system which in turn could lead to a weakness for some disciplines.
MATERIALS AND METHODS
Line 89 – this should be outlined as criteria for entry into the study.
Line 100 – where is Table 1A?
2.2 Data Collection – were the horses all led from the same side? I understand the loose rein is trying to eliminate handler error, but it would have been better if the horses had been led from both sides with a loose rein. In the placement of the inertial measurement units – the authors mention ‘pelvis (sacrum)’ – be more specific e.g. pelvis sacral tuberosity’s (left and right) or sacral dorsal spinous processes.
2.3 Data Processing – this needs to be clarified further – as there are sensors on the poll and pelvis, I am assuming you are targeting forelimb and hind limb asymmetry. How do you rule out lameness needs to be more specific assuming you identify this by mean displacement variable ≥ 25mm. There just needs to be more clarity in the definitions.
Were the tuber coxae and or sacral tuberosities identified in each horse as symmetrical or asymmetrical prior to the study?
RESULTS
Here a table identifying the asymmetric limb gait to the racetrack would have been beneficial.
DISCUSSIONS
First and second paragraphs are more suited to the Introduction.
Line 261 – this would be, in my opinion, the first sentence for the discussion.
Line 282 – are you referring to the outside hind limb here? If so this aligns with the small femoral study on Thoroughbred racehorses that race and train in a clockwise direction?
Line 297 – you have not differentiated the difference between a trot gait to the gallop, and the following sentences seem to be in contrast to the results - Line 205 in results - For all significant differences, HKJC mean values were negative (left asymmetry), and STC mean values were positive (right asymmetry) (Figure 3). Furthermore, at the gallop the Thoroughbred racehorse changes lead strides post curvature upon entering the straight.
Line 316 – camber and curvature have been discussed in a previous study on the outside hind limb in Thoroughbred racehorses noting that increased biomechanical stresses where greater in the left hind when racing clockwise. Are you disputing this finding?
Line 325 – musculoskeletal adaptation to directional training/racing was previously described in a study and even nominated the muscles involved undergoing adaptation.
CONCLUSION
The ‘outside limb’ needs to be identified as to which limb – does this apply to both the fore and hind limb?
REFERENCING
Just make sure it complies with the format from ‘Animals’ in the referencing section.
Reviewer 3 Report
Comments and Suggestions for Authors
This study compares objectively measured movement asymmetries in Thoroughbred racehorses trained and raced under different conditions. The reasons for performing the study and the objectives are clearly described. Some of the methodology requires further explanation and the presentation of the Results could be improved to facilitate readability. The Discussion requires radical revision to avoid repetition from earlier in the manuscript and to focus on discussion of the results of the study. In many places the text could be simplified to make it easier to read. The use of the term cohorts without definition is confusing.
Line 73 Knowledge
Materials & Methods
It would be useful to include more information about the layout (shape & dimensions & banking) of the training and racing tracks and the surfaces on which the horses trained and the maintenance of those surfaces - all of which could potentially be significant factors in the outcome of the study.
In addition it might be helpful to include information about daily training practices - duration of slow & fast work, for example; did all horses do fast work every day, for example?
Over what time period were the data collected?
Lines 95- 102 Were the data normally distributed? if not please present median & IQR & range; if normally distributed please use the scientific term mean
This information is really Results
Line 98 Were there any significant differences in age between the 2 horse populations?
Line 102 Remove 'As for shoeing ' Is 'primarily' necessary or the right word?
Line 307 It would make much more sense to start by stating that your exclusion criteria were x and from an initial cohort of y horses k were selected for study and then present the demographics of the population that were actually included in the study.
Line 116 Were all horses led from the left side? What head gear were the horses wearing? Was this consistent among all horses? How was speed controlled? At what stage in the day were the horses examined relative to their ridden work schedules? Was this consistent among all horses? Did any horses have a recent history of lameness that precluded training or required treatment? Were they all in daily training at the time of their evaluation?
Line 123 Do you actually mean the sacrum or the tubera sacrale?
Line 141 and 2 what is meant by 'unloading' in this context?
Line 144 Why were horses with no asymmetry (< 5mm) excluded from analysis? It would be helpful to state how many horses were symmetrical based on this definition (in the Results). Table 1 included 307 horses so obviously includes these horses which is a bit confusing!
It is not at all clear from the Results at what stage these horses were removed from the results presented
You have already explained that horses with considerable asymmetry (>25mm) were not included
Line 147 Include method of assessment of normality of data distribution in the M&M
Please define what is meant by cohort - the Results get confusing! – or just stick to HKJC & SJC
Results
Line 158 See previous comment about only including the 307 horses which were analysed
Line 163 & elsewhere. Any Table or figure legend should be able to be read independently from the text and therefore should provide more information. All abbreviations must be defined.
Line 168 - so is this for 307 horses? or 307 minus x horses with asymmetry < 5 mm? How many horses had asymmetry < 5 mm & why were they excluded?
Line 173 'Similar Chi squared tests of the pelvis related variables did not reveal any significant differences (p > 0.05 ' - seems to conflict with abstract
Line 40
Pelvis MinDiff (p=0.010) and UpDiff (p=0.021), and head MinDiff (p=0.006) and UpDiff (p=0.017) values were significantly different between cohorts
I think that the confusion arises through use of the term cohorts which is poorly defined. It would be much easier to understand if you stuck to SJC & HKJC throughout
Why 'similar'?
Line 174 Why use the flowery term 'did not reveal' rather than just say there was not significant difference
This comment applies to many other parts of the text which could be written in a more simple way
Line 176 I think (Left-Right) should come directly after 'left and right asymmetries ' not after cohorts
Line 178 Why 'similar'
Line 183 See previous comment about Table & figure legends. All abbreviations must be defined
Line 189 All Tables - please make sure that legends are comprehensive & all abbreviations are defined
Line 199 & elsewhere in section 3.2 It is quite ridiculous and of no biological relevance to present these figures to 3 decimal places!
Please explain / discuss why there are differences in the results of the mixed model analyses versus the Chi Squared tests
The results as presented are rather confusing and need discussion.
Discussion
Line 222 It is usual convention that a discussion starts by relating the study results to the hypotheses and then discusses the results in relation to previously published data & clinical relevance
The Discussion is not the place for a general description of global training and racing conditions
Please revise the Discussion accordingly and focus much more strongly on the interpretation of the results of your study. Please avoid excessive repetition of either the reasons for performing your study, the methods or results.
Radical revision is required.
Comments on the Quality of English Language
The wording could in many places be simplified to aid readability
Round 2
Reviewer 3 Report
Comments and Suggestions for Authors
The revised version of this manuscript is substantially improved.
Comment to Editor - authors would like to add a few words to the abstract to make it clearer about the statistical analyses performed & the results. I would recommend that this should happen
The figure legends still need to be more comprehensive so that each can be evaluated independently from the text
As indicated previously, I believe that it would be infinitely preferable if you started by stating that your exclusion criteria were x and from an initial cohort of y horses k were selected for study and then present the demographics of the population that were actually included in the study.
The authors responded to my questions about training intensity and when relative to training the horses were evaluated, but this information should be in the Methods. Also please include information about the time period during which data were collected.
Please state that speed was not controlled.
Any scientific study should have sufficient information in the Methods so that the study could be replicated.
Line 668 Jockey & Singapore are mis-spelt
The Discussion is now more relevant but is very repetitive. I have highlighted some of the repetitions but please re-read the manuscript carefully and remove all repetitions.
Line 672 ‘The present study was designed to investigate whether there are any predilections to how movement symmetry when trotting in a straight line in hand may adapt when horses are exercised in a predominant direction.’
This sentence as written is slightly misleading – this was not a longitudinal study – you do not know how these horses moved before initiating training so strictly speaking you cannot say that you were investigating adaptation.
Line 679 - 684 repetitive of Methods
Line 884 – The horses were not worked on a circle.
Lines 906-909 This is repetitive of the previous paragraph
Line 912 ‘it is plausible that horses racing and training in a predominant direction may also undergo changes to their symmetry of movement during straight line movement due to musculoskeletal adaptation to training and the results from this study support this’
This may be so – but since you did not assess the horses before the initiation of training this needs to b rephrased. You did not perform a longitudinal study.
Line 1000 ‘….when assessing the potential force being exhibited on the inside and outside during exercise in a circle’
Force is not exhibited
Line 1001 The horses work on more of an oval than a circle
Line 1032 Merge to create one sentence or rephrase to say 'This suggests ..'
Line 1089 ‘It is also important to note that daily training my have differed between horses. ‘ this is repetitive of line1023
Line 1097 ‘Allowing for musculoskeletal changes that may be important for the adaptive process for training and racing in a predominant direction have the potential to reduce injury rates in racing Thoroughbreds and should therefore be considered when training horses that intend to compete in a jurisdiction that races in the opposite direction to their usual way of going, as well as be taken into consideration when using objective measures of movement symmetry during objective symmetry assessments of Thoroughbred horses [46].’
What exactly do you mean by this? Specifically what are your recommendations for reducing injury rates? If the outside limb is more at risk of moving asymmetrically how are you ging to interpret this information at a lameness examination? Is there any data available from either jurisdictions about injury prevalence?
This is also rather repetitive of what has already been said Line 983 ‘Knowing if and how these various influences affect a horse’s 982 measurable movement symmetry would be useful to take into consideration when using objective methods to assess a horse movement, for example whilst undertaking lameness examinations, for longitudinal monitoring of horses’ movement during a training regime or when screening horses for their suitability to participate in racing or other equestrian competitions’
